# OBJVanish: Prompt-Driven Generation of Physically Realizable 3D LiDAR-Invisible Objects

Bing Li [1]  Wuqi Wang [2]  Yanan Zhang [3]  Jingzheng Li [4]  Haigen Min [2]  Wei Feng [5]  Xingyu Zhao [6]  Jie Zhang [7]
Qing Guo [8 9 †]

## Abstract

LiDAR-based 3D object detectors are fundamental to autonomous driving, where missed detections pose severe safety risks. While adversarial attacks are crucial for evaluating the robustness of these detectors, existing point-level perturbation methods rarely cause complete object disappearance and prove difficult to implement in physical environments. We introduce OBJVanish, a prompt-driven text-to-3D adversarial generation framework that enables physically realizable attacks by generating 3D object models that are effectively invisible to LiDAR-based 3D object detectors. We first conduct a systematic empirical study of detection vulnerability in LiDAR-based 3D object detectors, revealing multi-object compositions as the dominant factor. Based on this analysis, the proposed framework iteratively refines text prompts—optimizing verbs, objects, and poses—to generate LiDAR-invisible pedestrian instances as representative vulnerable road users under physical constraints. To ensure realizability, the framework operates over a curated pool of representative real-world 3D object models and restricts generation to their valid combinations. Extensive experiments show that OBJVanish consistently evades six state-of-the-art (SOTA) LiDAR-based 3D object detectors in both simulation and real-world settings, exposing critical vulnerabilities in safety-critical detection systems.

---

[1]Nanyang Technological University, Singapore [2]School of Information Engineering, Chang'an University, China [3]Hefei University of Technology, China [4]Zhongguancun Laboratory, China [5]Tianjin University, China [6]Wuhan University, China [7]Agency for Science, Technology and Research (A*STAR), Singapore [8]VCIP, CS, Nankai University, China [9]NKIARI, Shenzhen Futian, China. Correspondence to: Qing Guo <tsingqguo@ieee.org>.

*Proceedings of the 43rd International Conference on Machine Learning*, Seoul, South Korea. PMLR 306, 2026. Copyright 2026 by the author(s).

## 1. Introduction

LiDAR-based 3D object detectors (Lang et al., 2019; Shi et al., 2019; 2020; Hu et al., 2022; Zhang et al., 2022; 2023; 2024) are fundamental to modern autonomous driving and robotics. Their ability to accurately perceive and localize surrounding objects is critical for navigation safety, obstacle avoidance, and decision-making. However, missed detections—especially of vulnerable road users such as pedestrians—can lead to catastrophic outcomes. As these detectors are deployed in safety-critical settings, rigorously evaluating their robustness ensures practical reliability.

Adversarial attacks offer an effective way to probe such weaknesses before they are exploited in practice. Existing LiDAR-based attacks typically fall into three categories: sensor-level spoofing with external hardware (Zhu et al., 2021; Sato et al., 2024; Cao et al., 2023), digital perturbations to point clouds (Hamdi et al., 2020; Wang et al., 2021; Liu et al., 2023a), and mesh-based adversarial object insertions (Xiao et al., 2019; Cao et al., 2019b; Tu et al., 2020). While these approaches can degrade performance in controlled settings, they suffer from two key limitations: ❶ they rarely induce complete object disappearance, and ❷ they are often difficult to implement in real-world physical environments. Importantly, these methods operate on captured sensor data or handcrafted geometry, leaving the adversarial potential of generated 3D content unexplored.

In this work, we explore a different threat model: leveraging text-to-3D generative models to create physically grounded and semantically plausible 3D content that evades LiDAR-based 3D object detection. As such models improve in controllability and realism, an attacker can manipulate prompt semantics—without modifying sensor data—to generate adversarial 3D objects that remain effective in both simulation and physical settings. This raises a critical question: *Can we optimize textual prompts to generate physically realizable 3D objects that fool LiDAR-based 3D object detectors?*

To answer this question, we first conduct a systematic study of factors affecting the detectability of 3D pedestrians in simulated LiDAR environments. By manipulating single-object attributes (*e.g.*, topology, connectivity, and intensity), multi-

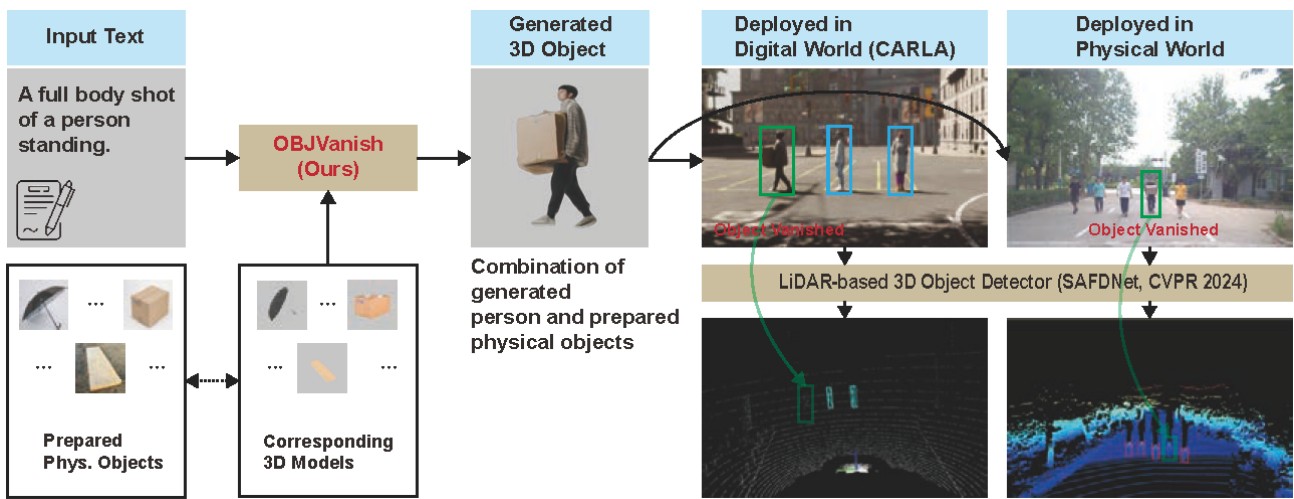

*Figure 1.* Example of **OBJVanish** for digital and physical deployment. OBJVanish generates physically realizable adversarial 3D human–object compositions from discrete text prompts that evade LiDAR detection ("Object Vanished"). For physical realization, we construct an object pool and instantiate the selected 3D models in the real world, enabling low-cost deployment by assembling objects without custom fabrication.

object compositions, and scene-level factors (*e.g.*, distance, viewpoint, and vehicle speed) in CARLA (Dosovitskiy et al., 2017), we find that *multi-object compositions dominate attack effectiveness and transferability to detectors*.

Building on this insight, we propose **OBJVanish**, a physically realizable text-to-3D adversarial generation framework that optimizes discrete prompts (verbs, objects, and poses) to synthesize LiDAR-invisible human–object compositions. OBJVanish is evaluated under both simulated and real-world settings, and further validated across diverse LiDAR configurations, including variations in beam density, motion patterns, and environmental conditions. The optimized prompts are fed into a Gaussian Splatting-based generator (*e.g.*, LGM (Tang et al., 2024)), producing point-based representations that are differentially rendered into LiDAR point clouds. To support real-world deployment, generation is constrained to a curated pool of physical objects, and multi-view renderings are used to guide accurate pose reproduction. Extensive simulation and physical experiments demonstrate that OBJVanish consistently fools multiple state-of-the-art LiDAR-based 3D object detectors (*e.g.*, SAFDNet (Zhang et al., 2024), HEDNet (Zhang et al., 2023), and PointRCNN (Shi et al., 2019)), exposing a practical and previously underexplored vulnerability in LiDAR-based 3D object detection.

Our contributions are summarized as follows:

- We present the first systematic analysis of 3D object detectability in simulated LiDAR scenes, showing that multi-object compositions have substantially stronger and more transferable effects than single-object attribute variations.

- We propose OBJVanish, a prompt-driven text-to-3D adversarial generation framework that optimizes discrete prompts to synthesize physically realizable LiDAR-invisible human–object compositions via a Gaussian Splatting-based pipeline.

- Extensive simulation and physical evaluations under realistic LiDAR variations demonstrate consistent evasion across multiple SOTA detectors, revealing a critical vulnerability to generative adversarial threats.

## 2. Related Work

### 2.1. Adversarial Attacks on LiDAR-based Detectors.

**Spoofing attacks.** These sensor-level attacks manipulate LiDAR outputs through external interference—such as projected lasers or reflected signals—without accessing the internal pipeline. They are typically categorized as injection attacks, which add fake points to mislead detectors (Cao et al., 2019a; Shin et al., 2017; Jin et al., 2023; Sun et al., 2020; Wang et al., 2023), and removal attacks (Cao et al., 2023; Sato et al., 2024), which eliminate real points to cause missed detections.

Spoofing attacks require specialized equipment and expertise, posing significant challenges for practical and reproducible physical deployment.

**Point cloud perturbation attacks.** Another line of research directly perturbs LiDAR point coordinates to perform adversarial attacks. For instance, (Wang et al., 2021) optimize a loss based on Intersection over Union (IoU), combined with imperceptibility constraints, to degrade the detection of car instances. To address detector non-differentiability, (Liu

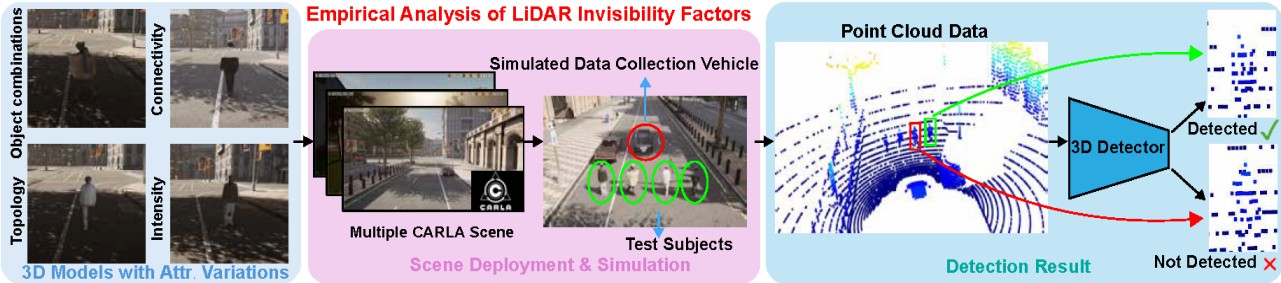

*Figure 2.* An overview of our empirical analysis pipeline for studying LiDAR invisibility factors. We manipulate the object attributes of 3D pedestrian models or human-object compositions, insert them into diverse CARLA scenes to collect data (middle), and finally evaluate their detectability using LiDAR-based 3D object detectors (right).

et al., 2023a) design a surrogate loss that balances minimal point changes with maximal disruption. ShapeAdv (Lee et al., 2020) introduces shape-aware adversarial attacks by perturbing the latent space of a point cloud auto-encoder to induce realistic geometric deformations that remain close to the original shape.

Despite their effectiveness in controlled settings, point cloud perturbation attacks require direct access to raw LiDAR points and fine-grained point-level control, limiting their practicality in real-world deployments.

**Adversarial object attacks.** A third line of research focuses on inserting physically realizable 3D objects into LiDAR scenes to perform adversarial attacks. A common strategy is to directly optimize mesh geometry or texture, often using differentiable LiDAR rendering (Tao et al., 2024) to enable gradient-based shape optimization. For example, (Cao et al., 2019b) employ differentiable ray casting to generate 3D-printable shapes that evade detection in real-world scenarios. Other works (Tu et al., 2020; Yang et al., 2021; Zheng et al., 2025; Xiao et al., 2019) insert predefined structures and refine their shape or appearance to interfere with nearby object detection.

While adversarial object attacks are amenable to physical deployment, they often rely on manually designed non-semantic meshes with limited scalability and stealth. In contrast, text-driven generative approaches enable scalable synthesis of semantically interpretable adversarial objects beyond manual geometric tuning.

### 2.2. 3D Content Generation and Prompt-Guided Attacks.

**3D generation paradigms.** Recent advances in 3D generation (Poole et al., 2022; Lin et al., 2023; Michel et al., 2022; Liu et al., 2023b; Xu et al., 2024; Shi et al., 2024; Team, 2024) have enabled high-quality 3D content synthesis from high-level inputs such as text or images, moving beyond sensor-level or manually crafted inputs. These pipelines typically combine prompt conditioning with multi-view 3D

reconstruction using implicit (*e.g.*, NeRF) or explicit (*e.g.*, Gaussian Splatting, triangle mesh) representations, producing editable 3D outputs with high controllability and realism. While primarily developed for benign use, such capabilities open new avenues for adversarial manipulation, where prompts can be optimized to synthesize physically realizable 3D objects that deceive perception systems.

**Adversarial attacks via 3D generation.** While prior works have explored 3D adversarial attacks using generative techniques, most rely on mesh-based representations. Following the seminal work in (Athalye et al., 2018), methods (Oslund et al., 2022; Suryanto et al., 2022; 2023; Wang et al., 2022; Zeng et al., 2019) have focused on texture-based perturbations, such as modifying vertex colors (Oslund et al., 2022; Xiao et al., 2019; Zeng et al., 2019) or optimizing texture maps on predefined meshes (Suryanto et al., 2022; 2023; Wang et al., 2022).

However, these approaches rely on photometric perturbations (*e.g.*, texture) that are inherently ineffective against geometry-sensing LiDAR, thus leaving the adversarial potential of geometric generation largely unexplored.

## 3. Empirical Analysis & Motivation

To thoroughly understand the vulnerabilities of LiDAR-based 3D object detectors, we first conduct an empirical study on the detectability of pedestrians in CARLA simulation environment. The test subjects are manipulated 3D pedestrians models, and we collect LiDAR data using a simulated car within CARLA, which allows for high-fidelity simulation across diverse and realistic conditions, including variations in weather, lighting, and scenes.

Prior studies have shown that geometric and physical properties—such as shape (Möller & Trumbore, 2005; Tu et al., 2020; Lee et al., 2020), structural coherence (Xiao et al., 2019), and intensity (Park et al., 2024)—can significantly impact LiDAR-based perception. Therefore, we focus on *modifications to individual objects, including changes in structure, appearance, and combinations of multiple objects.*

Specifically, we alter the structure by modifying the pedestrian's topology or adjusting its connectivity to simulate different types of occlusion. Additionally, we modify the pedestrian's appearance in LiDAR by adjusting its intensity. For object combinations, we simulate various human-object compositions, such as a pedestrian carrying a box or holding an umbrella. Visualizations of these attribute variations are provided in Appendix Fig. D(d-f). As shown in Fig. 2, our empirical study comprises three stages, providing a realistic assessment of how various factors affect the detection of generated objects.

Beyond object-level attributes, we also explore the *impact of scene-level factors, including the object's viewing angle, distance to the ego vehicle, and ego velocity*, on LiDAR detection outcomes. These variables reflect real-world driving scenarios and are crucial for understanding the robustness of detectors to adversarial attacks on generated objects in dynamic environments. We formalize the relationship between these factors and detection performance using the following decomposition:

$$\text{Detectability:} \quad \gamma = \mathcal{D}\big(\text{Obj}(\tilde{\text{Ped}} \cup \text{AddObj}), \\ \text{Env}(\mathcal{A}, \mathcal{D}, \mathcal{V})\big). \quad (1)$$

where $\gamma$ denotes the confidence score returned by the LiDAR-based 3D object detector $\mathcal{D}$ when evaluating the composite object instance $\text{Obj}(\tilde{\text{Ped}} \cup \text{AddObj})$ under the environment configuration $\text{Env}(\mathcal{A}, \mathcal{D}, \mathcal{V})$. Here, $\tilde{\text{Ped}}$ represents the generated pedestrian parameterized by topology $\mathcal{T}$, connectivity $\mathcal{C}$, and LiDAR intensity $\mathcal{I}$, while AddObj denotes optional auxiliary objects combined with the pedestrian. $\text{Env}(\mathcal{A}, \mathcal{D}, \mathcal{V})$ encodes the scene context, with $\mathcal{A}, \mathcal{D},$ and $\mathcal{V}$ representing the viewing angle, object distance, and ego velocity, respectively, all of which are controlled within the CARLA environment. This formulation captures how object attributes and scene conditions influence detectability.

Our quantitative results in Tab. 1 reveals important insights into how attribute modifications affect adversarial attacks on LiDAR-based 3D object detectors. ❶ Object combinations (AddObj) have the greatest impact on $\text{Obj}(\cdot)$ detectability $\gamma$, as combining multiple objects causes a more significant change in the target's point cloud distribution, thereby affecting LiDAR detection results. In Appendix A.1, we further study the effect of attaching AddObj at different spatial positions around the pedestrian. ❷ Topology ($\mathcal{T}$), Connectivity ($\mathcal{C}$), and Intensity ($\mathcal{I}$) have a moderate attack effect on LiDAR-based 3D object detectors, as changes to these attributes can alter the object point cloud distribution to some extent. However, these attribute variations do not generalize across detectors, with some detectors being unaffected by attribute variations (*e.g.*, Intensity).

We also gain several insights into the robustness of the attack effect of multiple object combinations under environmental factors, as shown in Fig. 3. ❶ The attack success rate varies significantly with distance ($\mathcal{D}$), indicating that this combinations are more effective at certain distances, challenging the attack's robustness across varying distances. ❷ Unlike simple point addition or removal, adversarially generated objects exhibit viewpoint and motion robustness due to the way $\text{Obj}(\cdot)$ responds to changes in $\mathcal{A}$ and $\mathcal{V}$.

# 4. Methodology

Motivated by our empirical analysis, we introduce OBJ-Vanish—a discrete prompt-based framework that generates physically realizable 3D objects capable of evading LiDAR-based 3D object detectors in Sec. 4.1 through an end-to-end optimization process over the structured verb-object-pose (VOP) prompt representation.

Moreover, to ensure real-world feasibility, we propose to curate a closed-set pool of physical objects to serve as adversarial props, which allows us to optimize the combination strategy of human subjects and these props for the adversarial object generation. As a result, we can physically implement the optimized combination strategy.

## 4.1. Physically-Informed Text-to-3D Adversarial Generation

**Overview.** We first build a verb-object-pose (VOP) textual prompt representation $\mathcal{P}$, which can interpret the target object we want to generate in terms of different verbs ($\mathcal{P}_v$), objects ($\mathcal{P}_o$), and poses ($\mathcal{P}_p$). Our method aims to find a specific textual prompt in $\mathcal{P}$, which can drive a pre-trained text-to-3D generation model $\mathcal{G}$ to generate an adversarial 3D object that the LiDAR-based 3D object detector cannot detect. To this end, we leverage the detection confidence from a surrogate detector as the adversarial objective function to guide the optimization in $\mathcal{P}$.

**Verb-object-pose (VOP) textual prompt representation.** We define a discrete VOP space comprising three interpretable subsets: verbs $\mathcal{P}_v$ (*e.g.*, hold"), objects $\mathcal{P}_o$ (*e.g.*, umbrella"), and poses $\mathcal{P}_p$ (*e.g.*, "on the head"). As detailed in Appendix A.2, this space consists of $|\mathcal{P}_v| = 7$ verbs, $|\mathcal{P}_o| = 13$ objects, and $|\mathcal{P}_p| = 7$ poses, yielding 637 semantic combinations that balance adversarial diversity with optimization efficiency. We sample a prompt from each subset and form a triplet, which can be represented as $(\varphi_v \in \mathcal{P}_v, \varphi_o \in \mathcal{P}_o, \varphi_p \in \mathcal{P}_p)$. Then, we can append the triplet to a base description $\varphi_{\text{base}}$ (*e.g.*, "a full body-shot of a person standing.") to form a complete prompt $\varphi \in \mathcal{P}$ (*e.g.*, "a full body-shot of a person standing and holding an umbrella on the head."):

$$\varphi = \text{concat}(\varphi_{\text{base}}, \varphi_v, \varphi_o, \varphi_p), \\ \text{s.t. } (\varphi_v, \varphi_o, \varphi_p) \in \mathcal{P}_v \times \mathcal{P}_o \times \mathcal{P}_p, \quad (2)$$

*Table 1.* Detection success rates (%) of different LiDAR-based 3D object detectors under various object attributes and scene configurations.

| | Single Pedestrian | | | Ped. w/ Obj. combinations | Point-based | | Voxel-based | | Point-voxel-based | |
|---|---|---|---|---|---|---|---|---|---|---|
| | Structure | | Appearance | | | | | | | |
| | Topology | Connectivity | Intensity | | PointRCNN | IA-SSD | SAFDNet | VoxelNeXt | HEDNet | PDV |
| **Scene 1** | ✓ | | | | 89.6 | 83.6 | 91.0 | 83.1 | 87.0 | 81.3 |
| | | ✓ | | | 52.1 | 23.4 | 83.6 | 82.9 | 83.3 | 81.0 |
| | | | | | 82.4 | 70.8 | 83.3 | 73.4 | 79.8 | 67.0 |
| | | | ✓ | | 73.6 | 72.9 | 84.3 | 73.1 | 75.7 | 69.5 |
| | | | | ✓ | **37.6** | **31.7** | **36.1** | **26.2** | **30.1** | **21.1** |
| **Scene 2** | ✓ | | | | 92.5 | 84.7 | 95.3 | 87.8 | 86.4 | 76.4 |
| | | ✓ | | | 51.3 | 15.9 | 90.2 | 87.6 | 84.1 | 75.8 |
| | | | | | 77.3 | 70.1 | 81.7 | 72.8 | 74.1 | 61.5 |
| | | | ✓ | | 74.2 | 64.0 | 89.4 | 72.0 | 73.5 | 61.4 |
| | | | | ✓ | **30.8** | **25.0** | **21.8** | **16.4** | **17.7** | **9.9** |
| **Scene 3** | ✓ | | | | 90.1 | 83.2 | 96.3 | 83.0 | 87.2 | 83.6 |
| | | ✓ | | | 47.7 | 26.1 | 88.7 | 83.0 | 84.8 | 83.6 |
| | | | | | 83.3 | 65.5 | 90.5 | 63.5 | 72.7 | 66.0 |
| | | | ✓ | | 71.2 | 61.6 | 87.0 | 61.4 | 68.3 | 61.6 |
| | | | | ✓ | **36.4** | **36.2** | **32.0** | **21.8** | **18.7** | **11.4** |

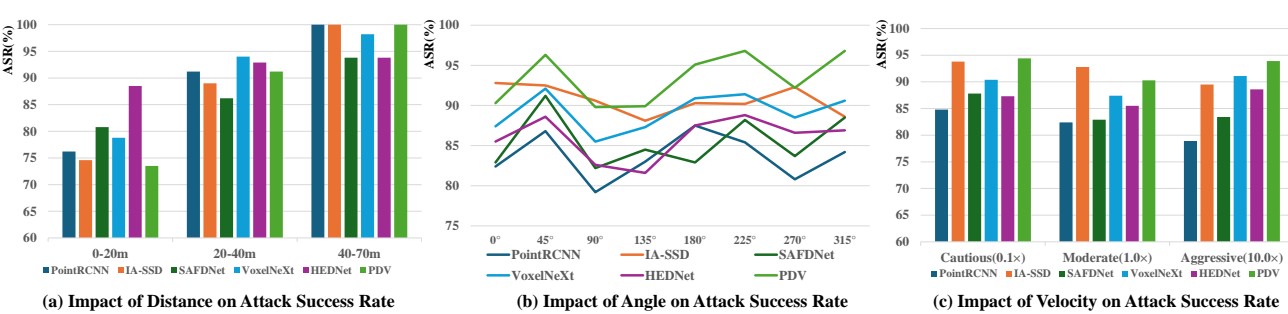

**(a) Impact of Distance on Attack Success Rate**     **(b) Impact of Angle on Attack Success Rate**     **(c) Impact of Velocity on Attack Success Rate**

*Figure 3.* Effects of environmental factors on 3D adversarial object detectability.

where concat denotes text-level concatenation of natural language components. We can feed the resulting prompt $\varphi$ to a text-to-3D model $\mathcal{G}$ to generate a 3D human-object instance. The key problem is how to conduct an effective optimization in the space $\mathcal{P}$ that depends on the three subsets to find a $\varphi$, with which the generated 3D object can mislead 3D object detectors. In the following, we introduce the optimization method and the corresponding formulated objective function.

**Detector-aware adversarial objective.** To find an effective prompt in the space $\mathcal{P}$, we conduct end-to-end optimization guided by a differentiable surrogate LiDAR-based 3D object detector. Specifically, we propose the following objective:

$$\varphi^* = \arg\min_{\varphi \in \mathcal{P}} \quad \mathcal{L}_{\text{adv}}\big(\mathcal{D}(\mathbf{P}), \mathcal{Y}\big)$$
$$\text{s.t.} \quad \mathbf{P} = \mathcal{S}\big(\mathcal{G}(\varphi), \varepsilon\big). \tag{3}$$

where $\mathcal{G}(\cdot)$ is a text-to-3D generation model, and $\mathcal{G}(\varphi)$ is to generate a 3D object according to the prompt $\varphi$. $\mathcal{S}(\mathcal{G}(\varphi), \varepsilon)$ is to embed the generated object into the environment $\varepsilon$ and produces a 3D point cloud $\mathbf{P}$. Note that, the environment $\varepsilon$ is the 3D point cloud of an existing environment. $\mathcal{D}(\mathbf{P})$ is a LiDAR-based 3D object detector to detect the objects in the $\mathbf{P}$ and yields a set of predictions $\{(\mathbf{b}_i, c_i)\}$, where $\mathbf{b}_i$ is the bounding box of detected object and $c_i$ is the detection confidence. We also have $\mathcal{Y} = \{\mathbf{b}^*\}$ as the set of ground-truth bounding boxes. We define the adversarial loss function as

$$\mathcal{L}_{\text{adv}}(\mathcal{D}(\mathbf{P}), \mathcal{Y}) = \max\Big( \max_i \big\{ c_i \cdot \mathbb{1}_{\text{IoU}(\mathbf{b}_i, \mathbf{b}^*) > \delta} \big\},$$
$$\eta \Big), \tag{4}$$

where the indicator $\mathbb{1}_{\text{IoU}(\mathbf{b}_i, \mathbf{b}^*) > \delta}$ selects boxes with an IoU above a threshold $\delta$ (*e.g.*, 0.5). If no match exists or the confidence is low, the loss is clamped to a constant margin $\eta = 0.1$. This prevents unnecessary unrealistic deformation after a successful attack.

**Latent-space optimization.** Directly optimizing over the discrete and combinatorial prompt space $\mathcal{P}$ is computationally infeasible due to its large cardinality. To enable efficient and differentiable optimization, we instead operate in the latent space of the frozen text encoder used by the pre-trained text-to-3D generator.

Specifically, we introduce three learnable logit vectors, $\mathbf{w}_v \in \mathbb{R}^{|\mathcal{P}_v|}$, $\mathbf{w}_o \in \mathbb{R}^{|\mathcal{P}_o|}$, and $\mathbf{w}_p \in \mathbb{R}^{|\mathcal{P}_p|}$, which parameterize categorical distributions over the verb, object, and pose subsets, respectively. To allow gradient-based optimization over discrete prompt choices, we apply the

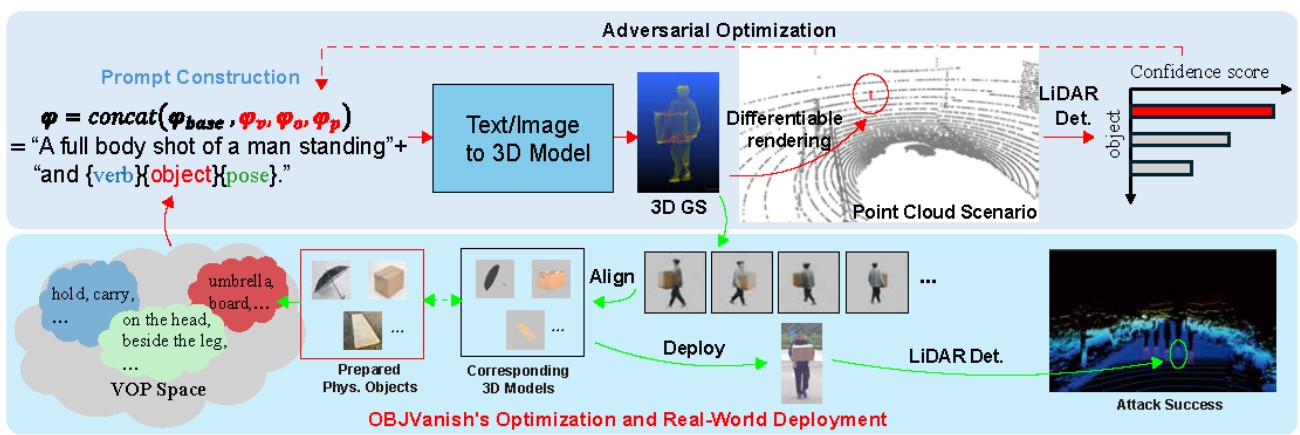

*Figure 4.* Overview of the OBJVanish framework: Prompt triplets (verb, object, pose) are optimized end-to-end by backpropagating an adversarial loss from a surrogate LiDAR-based 3D object detector. These prompts guide a text-to-3D model to generate human–object compositions, which are rendered into LiDAR scenes using differentiable rendering. Successful adversarial configurations are physically aligned with real LiDAR sensors for real-world validation.

Gumbel-Softmax relaxation (Jang et al., 2016).

Let $\mathrm{Enc}(\cdot)$ denote the frozen text encoder. For each component $x \in \{v, o, p\}$, we construct its latent representation:

$$\mathbf{E}_x = \sum_{i=1}^{|\mathcal{P}_x|} \pi_x[i]\, \mathbf{E}_{x_i}, \quad \pi_x = \sigma_{\mathrm{Gumbel}}(\mathbf{w}_x), \qquad (5)$$

where $\mathbf{E}_{x_i} = \mathrm{Enc}(x_i)$ denotes the encoder embedding of the $i$-th candidate in $\mathcal{P}_x$.

The final prompt embedding $\mathbf{E}$ is obtained by composing the base description with the selected verb, object, and pose components through the text-conditioning mechanism of the encoder. In practice, this composition is implemented by concatenating the corresponding textual components and aggregating their token-level embeddings produced by the frozen text encoder. The resulting embedding $\mathbf{E}$ is then provided to the text-to-3D generator $\mathcal{G}$ to synthesize a 3D object.

With this formulation, the detector-aware adversarial objective in Eq. (3) can be rewritten as

$$(\mathbf{w}_v^*, \mathbf{w}_o^*, \mathbf{w}_p^*) = \arg \min_{(\mathbf{w}_v, \mathbf{w}_o, \mathbf{w}_p)} \mathcal{L}_{\mathrm{adv}}\big(\mathcal{D}(\mathbf{P}), \mathcal{Y}\big),$$
$$\text{s.t.} \quad \mathbf{P} = \mathcal{S}\big(\mathcal{G}(\mathbf{E}), \varepsilon\big). \qquad (6)$$

where $\mathcal{D}$ denotes a differentiable surrogate LiDAR-based 3D object detector used during adversarial optimization. Optimizing $(\mathbf{w}_v, \mathbf{w}_o, \mathbf{w}_p)$ therefore corresponds to learning an effective verb–object–pose prompt that drives the generator to produce a 3D object with reduced detection confidence under LiDAR-based 3D object detectors.

### 4.2. Real-world Deployment Strategy

To ensure physical realizability without custom fabrication, we employ Geometric Instantiation to map the generated

adversarial results directly to the physical space. Our deployment strategy operates over a closed-set pool of existing physical objects, enabling direct real-world implementation. Given the closed-set object pool $\mathcal{O}_{\mathrm{phy}} = \{O_1, \ldots, O_N\}$, we first retrieve the optimal object index $k^*$ from the categorical weights $\mathbf{w}_o^*$ (Eq. 6) via:

$$k^* = \arg\max_{i \in \{1, \ldots, N\}} \mathbf{w}_o^*[i]. \qquad (7)$$

This discrete selection guarantees that the deployed adversarial object is always physically available.

Since the pose prompt optimization in Sec. 4.1 operates in the semantic space, we derive the precise spatial configuration by aligning the retrieved physical object $O_{k^*}$ with the generated adversarial mesh. This yields the rigid transformation matrix $\mathbf{T}^* \in SE(3)$:

$$\mathbf{T}^* = \begin{bmatrix} \mathcal{R}(\mathbf{r}^*) & \mathbf{t}^* \\ \mathbf{0}^\top & 1 \end{bmatrix}, \qquad (8)$$

where $(\mathbf{r}^*, \mathbf{t}^*)$ denotes the rotation and translation that minimize the geometric difference between the physical and generated objects.

Finally, the physical scene $S_{\mathrm{phy}}$ is assembled by applying $\mathbf{T}^*$ to the retrieved object $O_{k^*}$ relative to the pedestrian:

$$S_{\mathrm{phy}} = P_{\mathrm{ped}} \cup \{\mathcal{R}(\mathbf{r}^*) \cdot \mathbf{x} + \mathbf{t}^* \mid \mathbf{x} \in O_{k^*}\}. \qquad (9)$$

This construction ensures the physical geometry strictly aligns with the digital adversarial sample.

## 5. Experiment

### 5.1. Threat Model

Our framework adopts a white-box to black-box transfer attack setting, consistent with prior physical adversarial studies (Cao et al., 2021). During the generation

stage, the attacker is assumed to have access to a differentiable surrogate LiDAR-based 3D object detector, enabling gradient-based optimization over the semantic prompt space (verb–object–pose) to guide the text-to-3D generator toward adversarial geometry. After optimization, the attack transitions to a fully black-box deployment: the attacker has no access to the victim model's architecture, parameters, or inference outputs during deployment.

## 5.2. Setup

**Digital & Physical Setup.** We conduct digital experiments in CARLA using sensor configurations consistent with the KITTI (Geiger et al., 2012) dataset, collecting data from three maps. We generate 100 adversarial 3D objects—50 pedestrian instances with varied topology, connectivity, and intensity, and 50 with different object combinations—for detectability analysis. Physical experiments are performed in an autonomous driving test field with a stationary ego vehicle equipped with an RS-Ruby 128-beam LiDAR. Approximately 5,000 real-world point cloud frames are collected and annotated using SUSTech POINTs (Li et al., 2020) for detector fine-tuning. Five participants perform outdoor trials, with one carrying an adversarial object in the OBJVanish setting, and ten adversarial poses are evaluated as the subject walks toward the vehicle.

*Table 2.* Attack success rate (%) under different prompt strategies. All methods are optimized on PointRCNN and evaluated on the average transferability across six LiDAR-based detectors.

| Prompt Strategy | ASR (Avg.) |
|---|---|
| Random Prompt | $71.0 \pm 6.6$ |
| LatentPerturb | $78.0 \pm 11.2$ |
| OBJVanish (verb-only) | $62.8 \pm 4.7$ |
| OBJVanish (verb + object) | $77.3 \pm 8.8$ |
| OBJVanish | $87.7 \pm 7.9$ |

**Optimization Setup.** We optimize the logits of discrete prompt components (verb, object, pose) using Adam with a learning rate of 1e-4, while keeping the 3D generation model and LiDAR detector frozen. Optimization runs for 300 steps, minimizing the surrogate detector's classification confidence on the generated object. All training is performed on four NVIDIA L40 GPUs.

**Evaluated Detectors.** We evaluate six high-performing 3D object detection models from the OpenPCDet framework (Team, 2020), including point-based (PointRCNN (Shi et al., 2019), IA-SSD (Zhang et al., 2022)), voxel-based (SAFDNet (Zhang et al., 2024), VoxelNeXt (Chen et al., 2023)), and point-voxel-based methods (HEDNet (Zhang et al., 2023), PDV (Hu et al., 2022)). These models are pre-trained on the KITTI dataset and fine-tuned across both digital and physical scenarios.

## 5.3. Evaluation Protocol

We evaluate detection and attack effectiveness using Detection Success Rate (DSR) and Attack Success Rate (ASR). A detection is counted as successful if the predicted bounding box has an IoU exceeding 0.5, while an attack is considered successful if no box is detected or the IoU falls below 0.1.

## 5.4. Overall Effectiveness of Prompt-Driven Attacks

Tab. 2 summarizes the ASR under different prompt strategies. While LatentPerturb achieves relatively high ASR through continuous-space optimization, it lacks semantic interpretability and physical deployability. In contrast, OBJVanish optimizes prompts in a discrete and interpretable space, achieving the highest average ASR and strong cross-detector generalization. Optimizing only verbs yields limited gains, whereas jointly optimizing verbs and objects substantially improves ASR, highlighting the importance of object semantics. Detailed performance metrics for each specific detector are available in Appendix Tab. A(b).

## 5.5. Cross-Dataset Transferability

As shown in Tab. 3, adversarial objects optimized in CARLA exhibit strong zero-shot transferability to KITTI and nuScenes, without any re-optimization. The persistence of attack effectiveness under significant dataset shifts—including changes in scene geometry, LiDAR noise statistics, and annotation conventions—demonstrates that OBJVanish captures fundamental vulnerabilities of LiDAR-based 3D object detectors rather than overfitting to a particular dataset. For a granular analysis of transfer performance across individual detectors, please refer to the appendix.

*Table 3.* Cross-dataset Attack success rate (%) of OBJVanish. Optimized in CARLA and evaluated on KITTI/nuScenes models without fine-tuning.

| Dataset Transfer | ASR (Avg.) |
|---|---|
| CARLA → KITTI | 91.7 |
| CARLA → nuScenes | 94.0 |

**Comparison with SOTA Attack Methods.** Tab. 5 compares OBJVanish with representative adversarial baselines under the digital CARLA setup using the PointRCNN detector. The first two methods—AdvPC and NI-FGSM—perturb LiDAR point clouds directly, achieving moderate attack success rates of 62.9–64.2%, as the perturbations require manipulation of the vehicle's sensor point cloud data, making physical deployment challenging. To provide a more geometry-aware comparison, we reimplement (Tu et al., 2020)* and (Zheng et al., 2025)*, finding that their performance is more effective than fine-tuning the original point clouds. However, since they optimize the geometry of the 3D object, their optimization space remains limited. In con-

| LiDAR | LiDAR Setting | | | Trajectory types | | Weather Scenarios | | Velocity | |
| Detectors | 32-b | 64-b | **128-b** | Lateral | **Longitudinal** | Cloudy | **Sunny** | Fast | **Slow** |
|---|---|---|---|---|---|---|---|---|---|
| PointRCNN | 94.3 | 85.2 | 84.6 | 77.5 | 84.6 | 85.3 | 84.6 | 81.4 | 84.6 |
| IA-SSD | 96.2 | 95.0 | 90.3 | 80.9 | 90.3 | 92.8 | 90.3 | 91.5 | 90.3 |
| SAFDNet | 90.6 | 88.4 | 86.7 | 83.2 | 86.7 | 86.5 | 86.7 | 85.1 | 86.7 |
| VoxelNeXt | 95.0 | 91.1 | 90.6 | 78.2 | 90.6 | 88.9 | 90.6 | 90.3 | 90.6 |
| HEDNet | 92.5 | 88.3 | 85.3 | 80.8 | 85.3 | 82.8 | 85.3 | 86.4 | 85.3 |
| PDV | 93.0 | 85.7 | 81.6 | 72.0 | 81.6 | 83.5 | 81.6 | 82.2 | 81.6 |

*Table 4.* Attack Success Rate (%) of Different LiDAR-based Detectors under Various Physical Settings. Bold indicates the baseline configuration. -b refers to the number of beams in the LiDAR.

trast, OBJVanish optimizes discrete prompts to generate realistic and undetectable 3D human-object compositions, achieving a significantly higher ASR of 94.6%. Semantically rich prompts offer better interpretability, making real-world deployment easier, while providing greater optimization potential and enabling the generation of diverse object compositions that can impact LiDAR detection.

*Table 5.* Attack success rate (%) comparison against SOTA attack methods on PointRCNN (CARLA, digital setup).

| Method | ASR |
|---|---|
| AdvPC (Hamdi et al., 2020) | 64.2 |
| NI-FGSM (Lin et al., 2020) | 62.9 |
| (Tu et al., 2020)* | 79.3 |
| (Zheng et al., 2025)* | 68.4 |
| ScAR (Lu & Radha, 2023) | 47.5 |
| OBJVanish (Ours) | 94.6 |

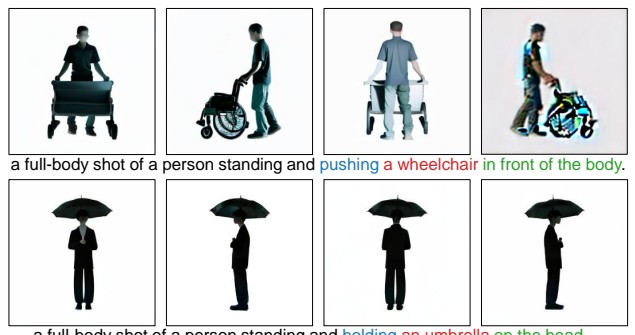

a full-body shot of a person standing and pushing a wheelchair in front of the body.

a full-body shot of a person standing and holding an umbrella on the head.

*Figure 5.* Multi-view renderings of an adversarial 3D object generated by OBJVanish, illustrating cross-view consistency and alignment with optimized semantic prompts.

**Physical Deployment Evaluation.** As shown in Tab. 4, ASR is sensitive to the number of LiDAR beams, with attack success decreasing as the beam count increases from 32-b to 128-b. The ASR drops the most for PDV, from 93.0% with 32-b to 81.6% with 128-b, indicating that fewer beams reduce detection capability, making the attack more effective. ASR also varies with motion type, with longitudinal motion achieving higher success rates than lateral motion, suggesting that lateral movement is more resistant

to attacks. Our method demonstrates robustness to weather and velocity changes, as evidenced by consistent ASR values of 85.3% in cloudy conditions and 84.6% in sunny conditions on PointRCNN, as well as 90.3% at fast velocities and 90.6% at slow velocities on VoxelNeXt. Some of our findings in real-world validation share similarities with those in simulation, such as the impact of environmental factors like vehicle speed and weather.

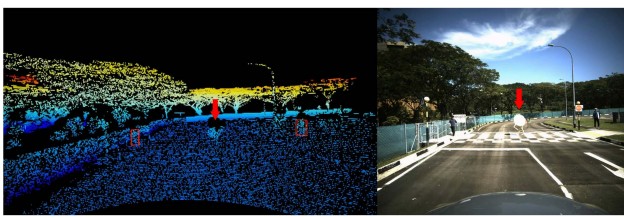

*Figure 6.* Visualization demonstrating the success of the OBJVanish attack on the SAFDNet LiDAR detector in a real-world deployment scenario.

**Visualization.** Fig. 5 demonstrates multi-view images of an adversarial object generated by OBJVanish, showcasing consistency across multiple views and a physically realizable multi-object composition based on the given prompts. We also present some failure cases in Appendix A.3. Additionally, Fig. 6 demonstrates the effectiveness of our physical deployment method, where the adversarial pedestrian walking on the zebra crossing was undetected by the detector. This real-world failure case highlights the practical risks posed by adversarially crafted objects and underscores the vulnerability of current LiDAR-based 3D object detectors.

## 6. Conclusion

We present the first empirical study on 3D object detectability in LiDAR-based detectors, revealing that single-object variations fail to generalize while multi-object compositions are more effective. Building on this insight, we propose OBJVanish, a physically realizable text-to-3D framework that exposes previously underexplored vulnerabilities through extensive simulation and real-world evaluations.

## Acknowledgment

This work is partly supported by the National Natural Science Foundation of China (62506111), the Fundamental Research Funds for the Central Universities (No. XXX-63263254), the China Postdoctoral Science Foundation (2025M781468), the Postdoctoral Fellowship Program of CPSF (GZC20251098).

## Impact statement

This study identifies a significant security gap in LiDAR-based 3D object detection, a cornerstone of safe autonomy. By showing that physically realizable human-object compositions can effectively bypass state-of-the-art detectors, we expose the inherent fragility of current perception pipelines.

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

# A. Appendix

In this section, we provide additional analyses, including the impact of body-part occlusions on LiDAR-based pedestrian detection, a discussion of our discrete prompt space for 3D generation, and visual comparisons of text-to-3D generation methods(*e.g.*, LGM, Meshy, and Shap-E). We also highlight limitations and future directions, focusing on spatial grounding, stochastic generation, and multi-sensor integration. For completeness, we additionally report full per-detector results for cross-dataset transferability in Table A(a) and prompt strategy comparisons in Table A(b).

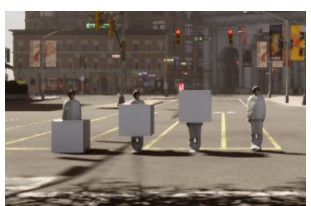

*Figure A.* Visualization of simulated body-part occlusions. From left to right: feet occlusion, torso occlusion, head occlusion, and unoccluded subject.

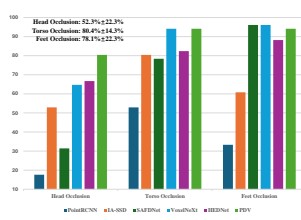

*Figure B.* Attack success rates (%) for different body-part occlusions across six LiDAR-based 3D detectors. Bold numbers (top-left) indicate mean and standard deviation per occlusion type.

## A.1. Occlusion Analysis by Body Region

As an extension of our empirical study, we report an additional occlusion-based experiment that investigates how occlusions of specific body regions affect LiDAR-based pedestrian detection. As shown in Fig. A, we simulate selective occlusions by placing geometric cubes over the feet, torso, and head regions of human subjects in the CARLA environment, allowing us to isolate each region's contribution to 3D detector performance.

The quantitative results are shown in Fig. B, where the attack success rate is reported for each occlusion type across six state-of-the-art LiDAR-based detectors. The bolded statistics in the top-left corner summarize the average attack success rate and standard deviation across detectors for each occlusion region. Notably, torso occlusion leads to the highest average attack success rate (80.4%) with relatively low variance, indicating that the torso contains critical structural features essential for reliable detection. Feet occlusion also causes significant degradation (78.1% average success rate), likely due to the loss of ground-contact cues in point clouds. In contrast, head occlusion results in much lower attack success (52.3%) and higher variability, suggesting that the head contributes less to LiDAR-based detection, and detectors are more robust to its absence.

These findings highlight the unequal contributions of body regions to 3D perception, with the torso and feet being

*Figure C.* Visualization of the discrete prompt space and corresponding 3D outputs. Left: Verb-object-pose (VOP) prompt pool. Right: Multi-view 3D generations illustrating compositional controllability from sampled triplets.

more crucial than the head. This motivates our use of spatially guided prompts to control object placement—favoring lower-body occlusions to improve adversarial stealth with minimal visual footprint.

## A.2. Discrete Prompt Space and 3D Generation

This section complements the main paper by providing additional visualizations and analysis of our discrete prompt space for controllable 3D generation. As shown in Fig. C, the left panel presents the verb-object-pose (VOP) prompt pool, where each prompt is formed by selecting one token from each of the three categories. This compositional design enables fine-grained control over human-object interactions and facilitates diverse, semantically meaningful generations. The right panel of Fig. C shows representative multi-view images generated from sampled triplets. Examples such as "holding an umbrella above the head" or "pushing a wheelchair in front of the body" demonstrate how semantic modularity leads to rich and plausible physical configurations. The discrete nature of this space further supports efficient exploration and adversarial prompt optimization, allowing us to discover configurations that are both physically realizable and evasive to LiDAR-based 3D object detectors.

Overall, these visualizations reinforce the main claim that compositional prompts enable interpretable and controllable adversarial generation, providing a flexible and scalable mechanism for producing diverse 3D attack candidates.

## A.3. Prompt-Conditioned Generation: Visual Comparisons and Failures

Beyond prompt design, we provide qualitative comparisons to explore how different text-to-3D backbones interpret prompt semantics, with a focus on generation consistency and common failure patterns.

We begin by visualizing the outputs of LGM, Meshy, and Shap-E under identical textual prompts, rendered within simulated urban scenes using the CARLA environment (Fig. D). Subfigure (g) shows representative LGM-generated samples conditioned on diverse prompts, demonstrating strong

*Table A.* Full quantitative results corresponding to the main-text tables.

*(a)* Cross-dataset attack success rate (%).

| Dataset Transfer | PointRCNN[†] | IA-SSD | SAFDNet | VoxelNeXt | HEDNet | PDV |
|---|---|---|---|---|---|---|
| CARLA → KITTI | 91.7 | 87.2 | 86.1 | 83.5 | 88.4 | 89.0 |
| CARLA → nuScenes | 94.0 | 90.2 | 85.1 | 87.8 | 90.4 | 91.6 |

*(b)* Attack success rate (%) under different prompt strategies.

| Prompt Strategy | PointRCNN[†] | IA-SSD | SAFDNet | VoxelNeXt | HEDNet | PDV | Mean |
|---|---|---|---|---|---|---|---|
| Random Prompt | 75.1 | 73.3 | 69.0 | 71.5 | 67.6 | 69.7 | $71.0 \pm 6.6$ |
| LatentPerturb | 81.7 | 80.7 | 71.3 | 78.1 | 77.3 | 78.1 | $78.0 \pm 11.2$ |
| OBJVanish (verb-only) | 66.7 | 64.0 | 61.3 | 60.0 | 62.6 | 61.7 | $62.8 \pm 4.7$ |
| OBJVanish (verb + object) | 82.1 | 74.0 | 80.4 | 75.4 | 76.7 | 75.0 | $77.3 \pm 8.8$ |
| OBJVanish | **92.1** | **87.5** | **86.8** | **82.7** | **87.9** | **89.1** | **$87.7 \pm 7.9$** |

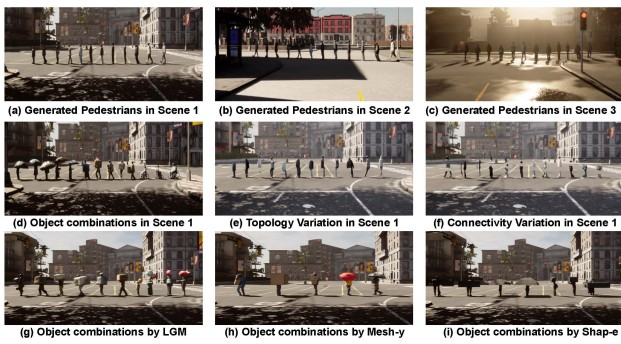

*Figure D.* Visualization of 3D pedestrians and attribute ablations in CARLA. Subfigures (a–c) depict 3D pedestrians across various scenes (Scene 1 - Scene 3), while (d–f) showcase object-level attribute ablations. (g) LGM results under diverse prompts, (h) Meshy-generated instances, and (i) Shap-E outputs.

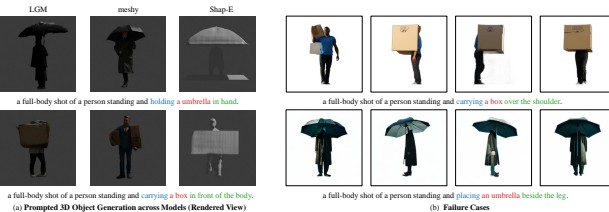

*Figure E.* Prompt-conditioned 3D generation and failure analysis. (a) Rendered views from LGM, Meshy, and Shap-E under identical prompts. (b) Failure cases from LGM: the top row shows inconsistent multi-view geometry; the bottom row shows spatial misalignment, where generated objects appear but are misplaced relative to the prompt instructions.

semantic alignment and consistent object placement. Subfigure (e) presents outputs from Meshy, a commercial closed-source model known for its high photorealism. Subfigure (f) displays results from Shap-E, which is characterized by greater geometric diversity but weaker adherence to prompt conditioning. These qualitative comparisons reveal characteristic trade-offs among the three models in terms of realism, diversity, and semantic controllability.

To further examine failure modes in prompt-conditioned generation, Fig. E(a) presents comparisons across LGM, Meshy, and Shap-E under identical prompts. The outputs differ notably in geometry, object placement, and overall fidelity. Meshy achieves high photorealism, but as a commercial closed-source model, it lacks controllability and transparency. Shap-E, while producing diverse structures, often yields implausible or structurally invalid results. In contrast, LGM offers a favorable trade-off between prompt adherence and visual realism, making it our primary choice for controlled generation.

Fig. E(b) highlights representative failure cases from LGM.

The top row shows inconsistencies across viewpoints, resulting in unrealistic multi-view geometry. The bottom row illustrates spatial grounding errors—for example, generating an umbrella far from the leg despite the prompt's instruction. These observations motivate future improvements in both spatial understanding and semantic controllability, even for the relatively more stable LGM.

### A.4. Model Limitations and Future Directions

Building on the above observations, we summarize key limitations of current text-to-3D pipelines. Despite the flexibility offered by compositional prompts, challenges persist in both semantic controllability and spatial grounding. First, while our framework searches over spatially meaningful prompts, existing backbones often struggle with executing complex spatial relations (e.g., placing an open umbrella in front of the body to obscure the legs), limiting precise grounding. Second, although verb-object-pose compositions support diverse generations, their effectiveness heavily depends on the semantic coverage of the pre-trained backbone—rare or unseen combinations can lead to implausible results. Third, the generation process is inherently stochastic; diffusion- or mesh-based pipelines may yield inconsistent outputs even

under identical prompts, impacting reliability. Finally, in the absence of explicit spatial supervision, achieving fine-grained physical realism remains a significant challenge.

Moreover, our current approach focuses solely on LiDAR-based detectors via geometry-level manipulations. However, modern autonomous systems typically integrate both Li-DAR and RGB cameras for robust 3D perception. Since our generated 3D assets also contain texture components, future work could explore jointly optimizing shape and appearance to fool both LiDAR and vision-based detectors. This direction may lead to more realistic, transferable, and comprehensive multi-sensor adversarial scenarios.

### A.5. The Use of Large Language Models

In this paper, we utilized a Large Language Model primarily for proofreading and refining the manuscript. The model assisted in improving language clarity, grammar, and overall writing quality. All content and ideas presented in this paper are the authors' own, and the LLM's role was limited to language enhancement.

### A.6. Full Quantitative Results

Appendix Tab. A(a) provides the full per-detector quantitative results corresponding to the cross-dataset transferability analysis reported in Table 3 of the main paper, where only the averaged attack success rates are presented for clarity. Appendix Tab. A(b) reports the complete per-detector attack success rates for different prompt strategies, serving as the detailed version of Table 2 in the main text, which summarizes results using mean and standard deviation.

