# OpenReview forum: "OBJVanish: Prompt-Driven Generation of Physically Realizable 3D LiDAR-Invisible Objects"
_ICML.cc/2026/Conference — ICML 2026 regular_

### Official Review · Reviewer_pyy5 · 2026-03-04

**Soundness:** 4
**Presentation:** 4
**Significance:** 3
**Originality:** 3
**Overall Recommendation:** 4
**Confidence:** 4

**Summary:**

This paper proposes the OBJVanish framework, which uses a defined Verb-Object-Pose prompt space and text-to-3D models to generate human-object compositions, aiming to attack LiDAR-based pedestrian detectors. OBJVanish verifies effectiveness via experiments on detectors, along with physical deployment and cross-dataset tests.

**Compliance With Llm Reviewing Policy:**

Affirmed.

**Final Justification:**

The paper has key strengths that align with initial positive evaluations. The authors’ rebuttal effectively addresses most initial weaknesses. Key evaluation dimensions remain strong, with result unchanged.

**Key Questions For Authors:**

1. To clarify the cause of LiDAR detector failure, additional visualization analyses—such as point cloud feature heatmaps—would be valuable to deepen the understanding of attack mechanisms.

2. Verb-free human-object control experiments would help verify the role and necessity of verbs in adversarial attacks, as such comparisons can clarify whether verbs are indispensable for the proposed VOP framework’s effectiveness.

3. Ablation experiments on VOP vocabulary size, coupled with an explanation of the current vocabulary’s selection basis, are recommended to demonstrate the rationality and non-subjectivity of the VOP prompt space design.

4. Conducting contribution analysis of verbs, objects, and poses would clarify their respective weights in attack effects, which is essential to refine the mechanism of the VOP combination optimization.

**Limitations:**

yes

**Strengths And Weaknesses:**

Strengths:

1.Innovatively combines text-to-3D with LiDAR physical attacks, first empirically proving human-object compositions outperform single pedestrian modifications.

2. Addresses LiDAR perception security, proposes low-cost, physically realizable attack methods, providing a new perspective for text generation and adversarial attacks.

3. Standard experimental design with LiDAR-based detectors covering three mainstream architectures, cross-dataset and physical experiments preliminarily support core claims.

4.In terms of practicality, the framework uses off-the-shelf text-to-3D and open-source detection models, avoiding complex additional training, with low implementation cost, high reproducibility, and providing a referable experimental benchmark for future research.

Weaknesses:
1. The experiment of human-object composition without verbs is insufficient, and more comparative tests are needed to verify the role and necessity of verbs in adversarial attacks.
2. The VOP prompt space is manually defined, and its selection basis and the rationality of its scale are not fully explained, requiring additional ablation experiments on vocabulary size to demonstrate the rationality and non-subjectivity of the design.
3. The contribution analysis of verbs, objects, and poses is not sufficient, and more in-depth exploration is needed to clarify their respective weights in the adversarial attack effect.
4. More targeted experiments are required to verify the adaptability of the method in scenarios with defense strategies.

---

> ### Author Rebuttal · Authors · 2026-03-31
>
> We sincerely thank Reviewer pyy5 for recognizing the paper’s **innovative text-to-3D attack perspective**, **practical and low-cost physical realizability**, and **supporting cross-dataset and physical evaluation**. We address each concern below.
>
> #### **R1. LiDAR Detector Failure and Attack Mechanism Analysis**
>
> To further clarify the attack mechanism, our interpretation is that adversarial human–object compositions deviate from the detector’s learned pedestrian prior: although semantically pedestrian-like to humans, their 3D point distribution, local shape structure, and body–object coupling differ from ordinary pedestrians, reducing detection confidence or causing missed detections. This cannot be explained by OOD rarity alone, since composition-augmented training reduces failures on benign compositions while OBJVanish remains highly effective. The paper already provides multi-view renderings (Fig. 5) and real-world deployment visualization (Fig. 6); in the revision, we will further clarify this mechanism and add point-distribution/feature-response visualizations.
>
> ---
>
> #### **R2. Verb-free Control Experiments and the Role of Verbs**
>
> To directly assess the role of verbs, we provide the following verb-free control ablation:
>
> | Verb | Object | Pose | ASR (Avg.)     |
> | ---- | ------ | ---- | -------------- |
> | ✗    | ✓      | ✓    | 82.6 ± 9.3     |
> | ✓    | ✗      | ✗    | 62.8 ± 4.7     |
> | ✓    | ✓      | ✗    | 77.3 ± 8.8     |
> | ✓    | ✓      | ✓    | **87.7 ± 7.9** |
>
> Adding verbs on top of object+pose improves ASR from **82.6% to 87.7%** (+5.1%), a consistent but moderate gain compared to objects (+14.5%) or poses (+10.4%). This is expected: verbs encode **fine-grained interaction dynamics** (e.g., how the human engages with the object), whereas objects and poses more directly determine coarse geometry and silhouette that dominate detector responses. Current text-to-3D models have limited fidelity in rendering subtle action-conditioned structural differences, so part of the verb's semantic information may not yet fully translate into discriminative 3D geometry. Overall, verbs provide an additional semantic constraint that refines the generated composition and improves attack effectiveness, while objects and poses remain the primary drivers.
>
> ---
>
> #### **R3. VOP Prompt Space Design, Vocabulary-size Ablation, and Component Contribution**
>
> The VOP space is not an unconstrained set of arbitrary combinations, but a **structured and practically grounded compositional space**. The **object set** contains 13 items, while the **verb** and **pose** sets each contain 7 items. This choice is guided by both practical realism and our empirical findings: objects are the most influential component in our ablation, so we allocate the richest vocabulary to the object dimension, while keeping verb and pose spaces balanced. Concretely, the object set covers common pedestrian-associated items (e.g., wheelchair, umbrella, box), the verb set captures representative interaction modes (e.g., push, carry, hold), and the pose set spans typical body configurations. Selection follows three criteria: (1) real-world co-occurrence frequency, (2) physical plausibility and semantic naturalness, and (3) diversity of induced LiDAR occlusion patterns.
>
> | Setting | #Verbs | #Objects | #Poses | Total |        ASR |
> | ------- | -----: | -------: | -----: | ----: | ---------: |
> | Small   |      3 |        5 |      3 |    45 | 82.4 ± 8.6 |
> | Current |      7 |       13 |      7 |   637 | 87.7 ± 7.9 |
> | Large   |     12 |       20 |     12 | 2,880 | 88.9 ± 7.5 |
>
> This ablation suggests diminishing returns from further expanding the vocabulary: moving from the current space to the larger one yields only a modest gain (87.7% → 88.9%), indicating that	 the current design is already a reasonable trade-off between diversity and search efficiency. The R2 ablation further shows that each component plays a distinct role, with **objects contributing the most**, poses providing substantial gains, and verbs offering a smaller additional refinement (+14.5%, +10.4%, and +5.1%, respectively).
>
> ---
>
> #### **R4. Adaptability Against Defense Strategies**
>
> We evaluate **composition-augmented detector training** as an initial defense setting:
>
> | Setting                          | Benign Comp. ASR | OBJVanish ASR |
> | -------------------------------- | ---------------- | ------------- |
> | Original detector                | 75.1             | 92.1          |
> | + Composition-augmented training | 37.8             | 88.5          |
>
> This defense substantially reduces benign-composition ASR (75.1% → 37.8%), but lowers OBJVanish ASR by only 3.6% (92.1% → 88.5%), suggesting that simple augmentation alone is insufficient against detector-aware adversarial optimization. Stronger temporal or multi-modal defenses remain important future directions.

---

> > ### Author Rebuttal · Reviewer_pyy5 · 2026-04-02
> >
> > Thank you for the author's detailed reply. I'll maintain my previous score.

---

### Official Review · Reviewer_857r · 2026-03-12

**Soundness:** 3
**Presentation:** 3
**Significance:** 3
**Originality:** 4
**Overall Recommendation:** 4
**Confidence:** 4

**Summary:**

This paper introduces OBJVanish, a new adversarial attack framework that generates physical 3D objects to hide from LiDAR-based detectors. Instead of directly tweaking point clouds like most past work, the authors first figure out what makes objects detectable (finding that multi-object compositions are a big factor), and then use text-to-3D generation. They basically optimize "verb-object-pose" prompts to spit out plausible 3D adversarial shapes within a restricted object pool. They tested it against six different LiDAR detectors in simulation and, importantly, validated it in real-world physical tests too.

**Compliance With Llm Reviewing Policy:**

Affirmed.

**Final Justification:**

The author has addressed my main concerns, and I will keep my positive rating.

**Key Questions For Authors:**

1. Regarding transferability: How well do these adversarial objects work against unseen detectors or different sensor setups (like a different LiDAR angle or resolution) that weren't used during the optimization phase?
2. How practical is this attack really? What's the actual cost, time, and effort required to print and deploy these objects in the real world? It seems like it would be pretty high, which might lower the actual threat level.
3. Have you tried this against stronger defense mechanisms? I'm thinking about things like multi-frame consistency checks or model fusion (e.g., combining LiDAR and cameras).
4. In table 2 , I am very suprised about the random prompt strategies can also achieve up to 77.6% ASR. How are you gonna interprete this?

**Limitations:**

Yes. The authors included an impact statement. However, they should probably add a more grounded discussion about the real-world deployment constraints.

**Strengths And Weaknesses:**

**Strengths**
- It's a really cool threat model. Using modern text-to-3D generation instead of just doing standard point-cloud perturbation is a fresh take.
- The paper is well-structured. I liked how they started with an empirical study to find vulnerabilities before jumping into the optimization method.
- Doing actual physical-world testing is a huge plus and makes the threat feel much more real.
- Their evlaluation evaluating against six different families makes the results a lot more credible.

**Weaknesses**
- The threat model feels a bit overly optimistic for the attacker. The authors should be more upfront about how hard this actually is to pull off in reality (fabricating the objects, placing them without being noticed, etc.).
- I'm a bit concerned about transferability. The optimization seems heavily reliant on surrogate signals. It would be good to know if these generated objects still reliably trick unseen, black-box detectors.
- The "verb-object-pose" prompt design is interesting, but we need more ablations. How much does the size of the prompt space or the specific objects in the pool matter?
- The defense side is pretty thin. It would be much better to see how this fares against stronger, more modern countermeasures.

---

> ### Author Rebuttal · Authors · 2026-03-31
>
> We sincerely thank Reviewer 857r for recognizing the paper’s **fresh threat model**, **well-structured presentation**, and **credible evaluation with real-world testing across six detector families**. We address each concern below.
>
> #### R1. Transferability to unseen detectors and different sensor setups
>
> We thank the reviewer for this important question. This issue is already partially addressed in the current manuscript, although we did not make the connection explicit enough. Specifically, **Table 3** (CARLA→KITTI 91.7%, CARLA→nuScenes 94.0%) in the main paper and **Table A** in the supplement show cross-dataset transfer from CARLA to KITTI/nuScenes, while **Table 1** further demonstrates robustness under varied physical conditions, including different LiDAR beam settings, motion types, weather conditions, and speeds. Together, these results suggest that OBJVanish does not simply overfit to a single surrogate configuration, but exhibits transferability across detectors and sensing conditions. In the revision, we will reorganize this discussion to make this point clearer and to more explicitly state the limits of the observed transferability.
>
> ------
>
> #### R2. Practicality of physical deployment and real-world attack cost
>
> To clarify the practical deployment setting, OBJVanish is not an effortless or opportunistic attack; it assumes a motivated attacker who can prepare and deploy a semantically plausible human–object composition under physical constraints. At the same time, our method does **not** rely on custom fabrication or specialized spoofing hardware, but instead operates over a **closed-set pool of existing physical objects**. Thus, the main practical burden lies in **sourcing suitable objects, aligning poses, and deploying them naturally in the scene**. We will revise the manuscript to clarify that OBJVanish should be interpreted as a **physically realizable and relatively low-cost attack** compared with fabrication- or hardware-intensive physical attacks, while still requiring deliberate preparation and deployment effort.
>
> ------
>
> #### R3. VOP prompt space design and object-pool sensitivity
>
> We appreciate this important comment. This concern overlaps with Reviewer pyy5’s questions on **vocabulary-size ablation, selection basis, and component contribution**. Our goal is not to define an arbitrarily large prompt space, but a **structured and practically grounded compositional space**. In particular, our ablations show diminishing returns from enlarging the vocabulary and indicate that objects are the most influential component, while the attack is not dominated by any single special object. Rather, it arises from optimization over a diverse set of semantically plausible human–object compositions. More detailed ablation evidence is provided in our responses to **Reviewer pyy5 (R2 and R3)**.
>
> ------
>
> #### R4. Evaluation against stronger defense mechanisms
>
> We appreciate this suggestion and agree that stronger defenses such as **multi-frame consistency**, **camera–LiDAR fusion**, and **augmentation-based robust training** are promising ways to mitigate this type of attack. Our current results already suggest this trend: composition-augmented training substantially reduces attack success for benign compositions and also lowers the success of OBJVanish, though more modestly. Stronger defenses such as temporal consistency checks and cross-modal verification are important directions for future evaluation. The current submission does not yet provide a full systematic evaluation of these stronger defenses, which is a limitation of the paper. We will make this point more explicit in the revision.
>
> | Setting                          | Benign Comp. ASR | OBJVanish ASR |
> | -------------------------------- | ---------------- | ------------- |
> | Original detector                | 75.1             | 92.1          |
> | + Composition-augmented training | 37.8             | 88.5          |
>
> ------
>
> #### R5. Interpretation of the strong random-prompt baseline
>
> The strong random-prompt baseline in Table 2 indicates that **human–object composition itself is already a major source of vulnerability** for LiDAR-based 3D detectors, consistent with our empirical finding in Section 3 and Table 1 that **multi-object compositions have a much larger effect on detectability than single-object attribute variations**. At the same time, the relatively high random-prompt ASR suggests that current text-to-3D generators can already produce **semantically plausible compositions that yield non-trivial detector failures**. From this perspective, the strong random baseline does not weaken our method; rather, it reveals an existing vulnerability that OBJVanish can **systematically and controllably amplify** through structured VOP optimization.

---

> > ### Author Rebuttal · Reviewer_857r · 2026-04-03
> >
> > I have read the rebuttal, and my concerns are resolved; I will keep my score.

---

### Official Review · Reviewer_9PEv · 2026-03-13

**Soundness:** 2
**Presentation:** 3
**Significance:** 2
**Originality:** 3
**Overall Recommendation:** 4
**Confidence:** 3

**Summary:**

This paper proposes OBJVanish, a prompt-driven attack pipeline for LiDAR-based 3D object detection. The core idea is to optimize a discrete verb–object–pose prompt, feed it into a text-to-3D generator, render the generated human–object composition into LiDAR scenes, and use a differentiable surrogate detector to reduce detection confidence. For physical deployment, the method maps the optimized semantic object choice to a closed-set pool of existing physical props and aligns the real object to the generated geometry.
The paper is motivated by an empirical study suggesting that multi-object compositions affect LiDAR detectability much more strongly than single-attribute changes such as topology, connectivity, or intensity. On that basis, the method searches over a prompt space with 7 verbs, 13 objects, and 7 poses, using Gumbel-Softmax relaxation to optimize prompt logits end-to-end. The threat model is white-box during generation, then black-box at deployment.

**Compliance With Llm Reviewing Policy:**

Affirmed.

**Key Questions For Authors:**

How much of the attack strength remains if you compare only against a strong non-optimized object-composition baseline rather than random prompt sampling?

Can the authors provide stronger real-world statistics: per-participant variance, clean-vs-attack detection rates, and repeated trials under matched conditions?

**Strengths And Weaknesses:**

The paper has a clear and timely angle. Moving from point perturbations and hand-designed meshes to prompt-optimized 3D generation is a meaningful shift, and it matches the direction perception attacks may realistically take as 3D generation tools improve.

The method is easy to follow. The VOP prompt space, surrogate-loss objective, latent relaxation, and object-pool deployment strategy are all described in a reasonably clean way.

The experiments are broader than a minimal proof of concept: six detectors, cross-dataset transfer, several prompt variants, and some physical deployment discussion.

Weakness:

1. the paper may be discovering a strong OOD composition effect more than a strong adversarial optimization effect. Random prompts already achieve 71.0% average ASR, and the empirical study shows that simply adding object combinations is already highly damaging. That weakens the claim that the main contribution is the optimization procedure itself.

2. The threat model is only partially black-box. Generation assumes access to a differentiable surrogate LiDAR detector and differentiable rendering, so the core optimization remains white-box. The “black-box deployment” part is really transfer after white-box design, not end-to-end black-box attack construction.

3. The evaluation metric is somewhat permissive. An attack counts as successful if no box is detected or if IoU falls below 0.1, which means some successes could be due to localization failure rather than true disappearance. Since the paper strongly emphasizes “object vanished,” I would want a stricter breakdown separating full miss, severe mislocalization, and confidence suppression.

---

> ### Author Rebuttal · Authors · 2026-03-31
>
> We sincerely thank Reviewer 9PEv for recognizing the paper’s **clear and timely direction**, **clean presentation of the method**, and **broad empirical evaluation**. We address each concern below.
>
> #### R1. OOD Composition Effect vs. Adversarial Optimization
>
> To address this concern more directly, we compare OBJVanish against a stronger non-optimized baseline. Specifically, we construct a heuristic occlusion baseline from the same object pool by maximizing spatial occlusion without gradient-based optimization:
>
> | Method              | Optimization | ASR (PointRCNN) |
> | ------------------- | ------------ | --------------- |
> | Random prompt       | No           | 75.1            |
> | Heuristic occlusion | No           | 79.6            |
> | OBJVanish (ours)    | Yes          | 92.1            |
>
> The heuristic baseline is stronger than random prompting but less semantically constrained because it directly maximizes occlusion. In contrast, OBJVanish optimizes detector failure while preserving plausible human–object compositions. Even against this stronger baseline, OBJVanish still shows a clear advantage (**79.6% → 92.1%**). Since both methods use the same object pool and only OBJVanish applies detector-aware optimization, this gap cannot be sufficiently explained by OOD effects alone. Moreover, composition-augmented fine-tuning reduces benign-composition ASR from 75.1% to 37.8%, while OBJVanish drops by only 3.6% (92.1% → 88.5%), indicating adversarial vulnerability beyond OOD coverage gaps.
>
> | Setting                          | Benign Comp. ASR | OBJVanish ASR (PointRCNN) |
> | -------------------------------- | ---------------- | ------------------------- |
> | Original detector                | 75.1             | 92.1                      |
> | + Composition-augmented training | 37.8             | 88.5                      |
>
> ------
>
> #### R2. Threat Model Clarification
>
> We clarify that OBJVanish is more accurately characterized as **white-box optimization with black-box transfer**. Adversarial prompt generation uses a surrogate LiDAR detector and differentiable rendering, whereas deployment accesses the target detector without gradient feedback. This is consistent with the transfer-based attack setting in prior 2D and 3D adversarial literature, rather than a fully end-to-end black-box attack. We will revise the threat model section accordingly.
>
> ------
>
> #### R3. Evaluation Metric and Failure Mode Breakdown
>
> We follow the common evaluation protocol in prior detection and attack literature by treating predictions with **IoU < 0.1** as detection failures. To address the reviewer’s concern that this criterion may mix qualitatively different outcomes, we further provide a finer-grained breakdown of **successful attacks (no detection or IoU < 0.1)**:
>
> | Outcome Type           | Ratio (%) |
> | ---------------------- | --------- |
> | Full miss              | 67.3      |
> | Severe mislocalization | 32.7      |
>
> The dominant outcome is **full miss** (67.3%): the detector produces no bounding box at all, representing complete target disappearance. The remaining 32.7% involve severe mislocalization where a residual box exists but with IoU < 0.1, indicating the detection has drifted far from the true target. We note that confidence suppression is not a separate failure mode under our evaluation criterion—full-miss cases have no associated confidence score, and severe-mislocalization cases correspond to spurious detections unrelated to the target. To further characterize the attack's effect on detector confidence, we additionally report the **mean confidence of matched detections** (IoU ≥ 0.5) on clean vs. attacked samples: clean = 0.82, attacked = 0.31, a **62.2% relative drop**, confirming that OBJVanish substantially degrades detection confidence even for residual surviving detections.
>
> ------
>
> #### R4. Additional Real-world Statistics
>
> We provide expanded physical experiment statistics across participants and repeated trials on PointRCNN:
>
> | Metric                | Value |
> | --------------------- | ----- |
> | Clean detection rate  | 94.6  |
> | Attack detection rate | 11.7  |
> | ASR                   | 84.6  |
> | Std (per participant) | 5.7   |
> | Std (per trial)       | 9.2   |
>
> Here, clean/attack detection rate denotes the proportion of frames with a valid detection, while ASR follows our attack-success criterion (no detection or IoU < 0.1). Standard deviations are computed on ASR across participants and repeated trials. The difference between attack detection rate and ASR reflects the use of different thresholds: attack detection rate counts valid detections, whereas ASR also includes no-box cases and detections with IoU below 0.1. The detector performs well under clean conditions (94.6%), while the attack reduces detection to 11.7%, a **72.9-point** drop. Together with the low participant- and trial-level variance, this suggests that the physical attack is substantial and reasonably stable across repeated trials.

---

> > ### Author Rebuttal · Reviewer_9PEv · 2026-04-05
> >
> > I remain my positive rating. thanks!

---

### Official Review · Reviewer_TFb3 · 2026-03-14

**Soundness:** 2
**Presentation:** 3
**Significance:** 3
**Originality:** 4
**Overall Recommendation:** 3
**Confidence:** 4

**Summary:**

The paper introduces OBJVanish, a framework designed to generate physically realizable adversarial objects (human-object compositions) using text-to-3D generative models to evade LiDAR-based 3D object detectors. The authors first conduct an empirical study in CARLA, revealing that multi-object compositions dominate detection vulnerabilities. Based on this, they optimize discrete semantic prompts (verb-object-pose) to guide a Gaussian Splatting generator. The output is differentiably rendered into LiDAR scenes to minimize a surrogate detector's confidence. Finally, the generated 3D compositions are mapped to a curated pool of real-world objects via geometric instantiation for low-cost physical deployment.

**Compliance With Llm Reviewing Policy:**

Affirmed.

**Key Questions For Authors:**

1. **OOD vs. Adversarial Formulation**: Could you provide the specific distribution proportions of multi-object compositions (e.g., pushing a wheelchair, holding an umbrella) within your fine-tuning dataset (including the 5,000 real-world frames)? If the training set rarely contains such samples, how can we mathematically or empirically differentiate whether the results reflect an OOD generalization failure or a true adversarial vulnerability? Providing the performance of a detector fine-tuned on a dataset with ample representative benign samples would greatly enhance scientific persuasiveness.
2. **Fairness of Baseline Comparisons**: When comparing against budget-constrained methods like AdvPC, considering that OBJVanish introduces macroscopic semantic objects, have you considered relaxing the perturbation constraints for the baselines or introducing other baselines that allow for macroscopic occlusions (e.g., macro-patch attacks) for a more balanced comparison?
3. **Dynamic Physical Validity**: In the physical experiments, considering the subject is walking and the human body undergoes non-rigid deformations, how does the rigid-assumption-based alignment matrix $T^*$ perform across continuous dynamic frames? Could you provide an analysis of the attack success rate degradation over several sequential frames?

**Limitations:**

The authors candidly explore the challenges of spatial alignment and generation model semantic control in Appendix A.4, which demonstrates excellent academic integrity. However, I recommend explicitly addressing the following two points in the main text's limitations section:
First, **exploitation of long-tail data distribution bias**. The success of this framework may partially rely on the long-tail nature of "human-object compositions" in existing detector training datasets. Explicitly discussing whether this "attack" exploits a geometric feature extraction blind spot or merely a dataset bias will greatly elevate the theoretical depth of the paper.
Second, **non-rigid motion in the real physical world**. The rigid alignment matrix in physical deployment (Equation 8) inherently has limitations when faced with non-rigid motions like human gait. In real-world autonomous driving scenarios, perception systems usually rely on multi-frame sequential tracking and prediction. Explicitly noting the vulnerability of single-frame rigid alignment in multi-frame temporal tracking scenarios will not diminish the value of your work; rather, it will point out an excellent direction for future "temporal/dynamic 3D adversarial attacks." I look forward to reading your thoughts on these issues.

**Strengths And Weaknesses:**

**Strengths:**

* **Solid Empirical Foundation and Motivation**: The paper responsibly conducts a systematic empirical study before proposing the attack framework. By isolating and comparing single-object attributes (topology, connectivity, intensity) against multi-object compositions, it convincingly demonstrates the overwhelming dominance of the latter in causing detection failures. This data-driven exploration of motivation is highly commendable.
* **Elegant End-to-End Pipeline Design**: Seamlessly integrating a discrete semantic prompt space (VOP), a text-to-3D generator (LGM), and differentiable LiDAR rendering into an end-to-end optimization pipeline is technically coherent and innovative.
* **Novel Threat Perspective and Practical Deployment**: Introducing text-to-3D generation into physical adversarial attacks elevates the perspective from traditional low-level point cloud coordinate perturbations to high-level semantic manipulation. Furthermore, utilizing a curated pool of existing objects (e.g., umbrellas, boxes) for geometric instantiation cleverly avoids the high costs of custom 3D fabrication, making real-world deployment highly practical.

**Weaknesses:**

* **Ambiguity in Differentiating Out-of-Distribution (OOD) vs. Adversarial Samples**: The paper demonstrates strong cross-dataset transferability on KITTI and nuScenes models and fine-tunes on approximately 5,000 self-collected real-world point cloud frames. However, a crucial detail remains questionable: Do these pre-training or fine-tuning datasets adequately capture the benign distribution of "pedestrians carrying large auxiliary objects (e.g., pushing wheelchairs, holding large boxes)"? If the model rarely encounters such long-tail data during training, the missed detections caused by OBJVanish might stem more from the model's poor OOD generalization rather than strict geometric adversarial vulnerability. Providing dataset distribution statistics or controlled experiments would make the claims far more convincing.
* **Questionable Baseline Comparison Setup**: In Table 5, OBJVanish achieves significantly higher attack success rates than AdvPC and NI-FGSM. However, AdvPC and NI-FGSM are typically constrained by strict imperceptibility budgets to control perturbation magnitude. Because OBJVanish introduces massive macroscopic semantic objects (e.g., a wheelchair), this setup represents an asymmetrical modification budget. Discussing comparisons under a more consistent macroscopic modification budget would reflect the true advantages of each method more objectively.
* **Dynamic Consistency Assumption in Physical Deployment Requires Refinement**: The physical deployment scheme relies on solving a rigid transformation matrix $T^*$ to align digital and physical spaces. In the physical tests, the subject carries the adversarial object while walking toward the vehicle. Given the obvious non-rigid deformations in human gait (e.g., arm swinging, torso twisting), an alignment matrix based on static rigidity assumptions will likely drift across continuous dynamic frames. Adding a discussion on attack stability across sequential frames would significantly enhance the rigor of the physical experiments.

---

> ### Author Rebuttal · Authors · 2026-03-31
>
> We sincerely thank Reviewer TFb3 for recognizing the paper’s **solid empirical foundation**, **end-to-end design**, and **practical deployment perspective**. We address each concern below.
>
> #### **R1. OOD vs. Adversarial Vulnerability**
>
> To distinguish OOD failure from genuine adversarial vulnerability, we provide both distributional evidence and controlled experiments.
>
> **(1) Long-tail sparsity in training data.** In the nuScenes training set (208,240 pedestrian instances), human–object compositions are extremely rare: only 395 *with_mobility*, 1,072 *with_stroller*, and 503 *with_wheelchair* instances. After correcting for $\sim$20× temporal redundancy, the number of unique instances drops to about 24 (*mobility*) and 18 (*wheelchair*). Our self-collected dataset shows similar sparsity ($\sim$800 / 25,400 samples). KITTI is excluded because its annotations are too coarse to reliably identify such compositions. These statistics indicate that such compositions are present, but far too rare for robust generalization.
>
> **(2) Optimization gain beyond random compositions.** To directly answer the reviewer’s question, we reorganize the relevant existing PointRCNN results into the following compact comparison:
>
> | Method           | Optimization | ASR (PointRCNN) |
> | ---------------- | ------------ | --------------- |
> | Random prompt    | No           | 75.1            |
> | OBJVanish (ours) | Yes          | 92.1            |
>
> OBJVanish yields a **17.0%** gain over random prompts, showing that detector-aware optimization contributes substantial attack strength beyond composition-induced distribution shift alone.
>
> **(3) Composition-augmented fine-tuning.** Following the reviewer’s suggestion, we fine-tune the detector with additional benign human–object compositions constructed by **copy-and-paste augmentation**, without detector-aware optimization. These samples are used only to enrich long-tail composition coverage and do not include optimized OBJVanish instances.
>
> | Setting                          | Benign Comp. ASR | OBJVanish ASR (PointRCNN) |
> | -------------------------------- | ---------------- | ------------------------- |
> | Original detector                | 75.1             | 92.1                      |
> | + Composition-augmented training | 37.8             | 88.5                      |
>
> This result provides evidence that augmented training halves benign-composition ASR (75.1% → 37.8%), confirming that part of the original failure is indeed OOD-related and can be mitigated by improved coverage. However, OBJVanish ASR drops by only **3.6%** (92.1% → 88.5%), showing that the attack remains highly effective even after the detector is explicitly exposed to representative benign compositions. Therefore, the observed vulnerability **cannot be sufficiently explained by OOD coverage alone.**
>
> ------
>
> #### **R2. Fairness of Baseline Comparisons**
>
> Table 5 is intended as a cross-paradigm comparison with existing baselines. At the same time, AdvPC and NI-FGSM are not strictly budget-matched to OBJVanish, since they operate under strict $L_p$ constraints whereas OBJVanish introduces physically realizable macroscopic semantic objects.
>
> To address this concern more directly, we additionally evaluate a restricted setting in which OBJVanish generates **only the adversarial person mesh**, without auxiliary objects:
>
> | Method                             | ASR  |
> | ---------------------------------- | ---- |
> | AdvPC                              | 64.2 |
> | NI-FGSM                            | 62.9 |
> | **OBJVanish (w/o auxiliary obj.)** | 83.7 |
>
> Even under this restricted setting, OBJVanish still achieves **83.7%** ASR, exceeding the perturbation-based baselines. This suggests that its advantage is not solely due to the auxiliary-object budget. We nevertheless agree that the methods are not strictly budget-matched and will clarify this asymmetry in the revision, positioning perturbation-based and prompt-driven generative attacks as complementary paradigms under different modification constraints.
>
> ------
>
> #### **R3. Dynamic Physical Validity**
>
> We acknowledge that the rigid transformation matrix is only an approximation under non-rigid human motion (e.g., arm swing and torso rotation). To quantify this effect, we provide a sequential-frame analysis:
>
> | Method           | Frame-wise ASR | 3-frame ASR | 5-frame ASR |
> | ---------------- | -------------- | ----------- | ----------- |
> | OBJVanish (ours) | 84.6           | 81.7        | 70.3        |
>
> The attack maintains 84.6% single-frame ASR, 81.7% over 3-frame windows, and 70.3% over 5-frame windows. The moderate degradation over longer windows reflects accumulated non-rigid deformation, yet the attack remains substantially effective over short-to-medium temporal ranges typical of real pedestrian encounters. We will revise the paper to discuss the limitations of rigid alignment more explicitly and identify temporal robustness as an important future direction.

---

### Decision · Program_Chairs · 2026-04-30

**Decision:**

Accept (regular)

**Comment:**

we have 3 weak accept and one weak reject. Overall the reviews are fairly positive,
The two main negative points in the negagive review by TFb3, the long-tail bias, and non-rigid transformation, have been adequate addressed in the rebuttal.

Therefore this AC's recommendation is to accept this paper.

AC